# Economic Responses of Maize, Soybean, and Wheat in Three Rotations under Conventional and Organic Systems

**William Cox** [1,*] , **John J. Hanchar** [2] , **Jerome Cherney** [1] and **Mark Sorrells** [3]

1   Unit of Soil and Crop Sciences, School of Integrated Plant Sciences, Cornell University, Ithaca, NY 14853, USA
2   Northwestern New York Dairy, Livestock, and Field Crops Program, College of Agriculture and Life Sciences, Cornell University, Ithaca, NY 14853, USA
3   Unit of Plant Breeding and Genetics, School of Integrated Plant Sciences, Cornell University, Ithaca, NY 14853, USA
*   Correspondence: wjc3@cornell.edu

**Abstract:** Low prices have prompted growers to contemplate transitioning to an organic system. We evaluated red clover-maize-soybean-wheat (Cl-M-S-W), maize-soybean (M-S-M-S), and soybean-wheat/red clover-maize-soybean (S-W/Cl-M-S) rotations in organic and conventional systems in New York, USA from 2015 to 2018 to identify profitable organic practices. Organic compared with conventional maize in 2017 had 14.6% higher yield and $2107/ha higher returns above selected costs in the S-W/Cl-M-S rotation; and had $1007/ha higher returns in the M-S-M-S rotation, despite 3.6% lower yield and higher production costs, because of the organic price premium. Likewise, organic compared with conventional soybean had ~$800 to ~$900/ha higher returns in 2017 and 2018, despite ~10% lower yield and ~$50/ha higher production costs, because of the organic price premium. Organic compared with conventional wheat yielded ~4% higher with $125/ha higher returns, despite ~$435/ha higher production costs. Organic compared with the conventional system had $1018/ha higher returns in the Cl-M-S-W rotation, $1782/ha higher in the M-S-M-S rotation, and $2961/ha higher in the S-W/Cl-M-S rotation in 2017 and 2018. Although returns in 2015 and 2016 (no organic premium) were lower, the organic compared with the conventional system from 2015 to 2018 had $673/ha higher returns in the Cl-M-S-W rotation, $497/ha higher in the M-S-M-S rotation, and $2355/ha higher in the S-W/Cl-M-S rotation indicating that the S-W/Cl-M-S rotation was the most profitable organic rotation during the four-year period.

**Keywords:** organic farming system; maize; soybean; wheat; returns

## 1. Introduction

Maize (*Zea mays* L.), soybean (*Glycine max* (L.) Merr.), and wheat (*Triticum aestivum* L.) prices have trended mostly downward since 2013 [1]. However, organic growers who produce maize, soybean, and wheat, receive significant price premiums. Consequently, some crop producers, who practice maize-soybean or maize-soybean-wheat/red clover (*Trifolium pratense* L.) rotations in the Northeast, USA, have contemplated transitioning from a conventional to organic cropping system in order to capture the organic premium. The United States Department of Agriculture (USDA), however, requires a 36-month transition period to reduce residual fertilizer and pesticide residue from the conventional farming system [2]. In addition, the use of most conventional inputs, including genetically modified (GM) crops, synthetic fertilizer, and pesticides are prohibited during the 36-month transition period before the land can be certified as organic and eligible for the organic price premium [2]. Furthermore, a USDA survey report indicated that organic compared with conventional maize, soybean or wheat

production had lower yields and higher production costs, which exacerbates the economic risk during the transition [3]. The identification of the best crop rotation, including the entry crop (first year transition crop), during the transition and the first two years after organic certification are essential for maintaining cash flow for crop producers who transition from a conventional to an organic cropping system.

An organic compared with a conventional maize-soybean rotation in a study in Minnesota, USA had mostly similar production costs but $692/ha lower net present value in the absence of the organic premium because of 34% lower maize and 15% lower soybean yields [4]. In the presence of an organic premium, however, the organic compared with the conventional maize-soybean rotation had $242/ha higher net present values. The authors reported that planting soybean instead of maize as the entry crop resulted in a $535/ha advantage of net present value in the organic maize-soybean rotation [5]. In another Minnesota, USA study from 1993 to 2010, an organic maize-soybean-oat (*Avena sativa* L.)/alfalfa-alfalfa (*Medicago sativa* L.) rotation compared with a conventional maize-soybean rotation had $86/ha lower production costs but $186/ha lower net revenue in the absence of an organic premium because of similar maize yields and 25% lower soybean yields [6]. In the presence of an organic premium, however, the four-year organic rotation had $483/ha higher net revenue.

An organic maize-soybean-wheat/red clover rotation compared with a no-till (NT) conventional maize-soybean rotation had $130/ha to $408/ha lower economic mean returns in the absence of organic premiums at two sites in Wisconsin, USA from 1993 to 2006 [7,8]. In the presence of government payments and organic premiums, the organic maize-soybean-wheat/red clover rotation had $165/ha to $321/ha higher economic mean returns. An organic maize-soybean-oat/alfalfa-alfalfa rotation compared with a conventional maize-soybean rotation in a study in Iowa, USA had higher profitability during the transition period (1999 to 2001) because of lower production costs and similar maize and soybean yields [9,10]. In the second phase of the study (2002 to 2010), the organic crop rotation was far more profitable because of lower production costs in maize, similar maize and soybean yields, and higher prices received for organic maize and soybean, associated with the organic premium [11].

Organic maize-soybean and organic maize-soybean-wheat rotations had similar mean returns ($585–$589/ha, respectively) compared to a conventional maize-soybean rotation using chisel tillage ($502/ha), but lower than a conventional NT maize-soybean rotation ($702/ha) in an economic analysis of a study in Maryland, USA from 2006 to 2014 [12]. Organic compared with conventional maize and wheat had higher production costs; whereas organic and conventional soybean had similar production costs. Organic maize yielded 69% of conventional maize, organic soybean yielded 72% of conventional soybean, and organic wheat yielded 92% of conventional wheat. Consequently, the organic price premiums were crucial for the two- and three-year organic rotations to maintain similar mean returns as the two-year conventional rotation.

Organic crop yields in farming system experiments are typically higher than yields on practicing organic farms [13], which is perhaps a reason why adoption of organic farming systems has been slow despite many studies showing higher profitability [14]. The probability of growers transitioning to organic maize, soybean, and wheat production increases when commodity prices are low [14]. In addition, a smaller farm size, typical of the Northeast compared to the Midwest, USA agriculture, increases the likelihood of adoption of organic cropping systems [14]. Consequently, we expect to see more growers adopting organic maize-soybean or organic maize-soybean-wheat/red clover rotations in the Northeast, USA. Two objectives of this study were: (a) to identify the best organic crop rotation that results in the best economic returns during the transition years, the first 2 years after organic certification, and the first 4 years of production and (b) to determine if high input management (high seeding and high N rates) can minimize weed competitiveness and improve soil N availability, thereby increasing yields and returns. We previously reported on the economics of three crop rotations, featuring maize, soybean, and wheat during the transition years in the absence of an organic premium [15]. This paper will focus on the economics of the three individual crops and the three crop rotations during the two years after the transition phase in the presence of an organic premium. We will then evaluate the

economics of the three crop rotations over the four-year period, which include the transition period before organic certification and the two years after organic certification.

## 2. Materials and Methods

We conducted our study from 2015 to 2018 at a Cornell University research farm near Aurora, NY, USA (42°44′ N, 76°40′ W) on three contiguous experimental fields (220 m × 40 m) that had a similar mixture of tile-drained silt loam soils (fine-loamy, mixed, mesic, Glossoboric Hapludalfs; and fine-loamy, mixed, mesic, Oxyaquic Hapludalfs). The three fields had different previous crops (spring barley (*Hordeum vulgare* L.), maize, and soybean) grown under conventional management practices in 2014. The three rotations evaluated from 2015 to 2018 included red clover (as a green manure crop)-maize-soybean-wheat (Cl-M-S-W), maize-soybean-maize-soybean (M-S-M-S), and soybean-wheat/red clover-maize-soybean (S-W/Cl-M-S) in conventional and organic farming systems (Table 1). The clover in the S-W/Cl-M-S rotation was frost-seeded into a standing wheat crop at the end of winter when the soil was still frozen. The 36-month transition period in the organic system consisted of only two growing seasons in our study because the previous crops before the transition did not receive any prohibited organic inputs (fertilizer, pesticides, etc.) after June of 2014. The 2017 crops were thus considered eligible for the organic price premium in the third year of our study because they were harvested in late September (soybean) and November (maize) in 2017, more than 36 months after harvest of the 2014 conventional crops.

**Table 1.** Sequence of the red clover-maize-soybean-wheat/red clover (Cl-M-S-W/Cl), maize-soybean (M-S-M-S), and soybean-wheat/red clover-maize-soybean (S-W/Cl-M-S) rotations from 2015 to 2018 at a Cornell University research farm in central New York, USA.

| Year | Crop Rotations | | |
|------|------------|---------|-----------|
| | **Cl-M-S-W/Cl** | **M-S-M-S** | **S-W/Cl-M-S** |
| 2015 | Red Clover (Cl) | Maize | Soybean |
| 2016 | Maize | Soybean | Wheat/RC |
| 2017 | Soybean | Maize | Maize |
| 2018 | Wheat/(Cl) | Soybean | Soybean |

The experimental design was a randomized complete block in a split-split-split plot arrangement with four replications. Previous crops (the three fields) were whole plots, farming systems (conventional and organic) were sub-plots, crop rotations were sub-subplots, and management inputs (recommended and high inputs) were sub-sub-subplots. The entire 40 m plot lengths were planted to maize, soybean or winter wheat at their appropriate planting dates in each field in their designated 3 m wide strips. Harvest plot length was shortened to 33 m to allow for 3.5 m borders on the north and south sides of the plots. Also, 3 m borders were inserted between sub-plots (farming systems) to minimize spray drift or fertilizer movement from conventional into organic plots. Likewise, 3 m border plots were inserted between each sub-subplot (crop rotations) to minimize border effects from each crop, which differed in height. Whole plot dimensions were 216 m wide and 33 m long, sub-plot dimensions were 27 m wide and 30 m long, sub-subplot dimensions were 6 m wide and 30 m long, and sub-sub-subplot dimensions were 3 m wide and 30 m long.

Table 2 provides an overview of the management practices for each crop in each farming system with different management inputs. Briefly, in the conventional rotations, we planted treated (insecticide/fungicide seed treatment) maize (GM hybrid), soybean (GM cultivar) and wheat cultivars and used synthetic starter and N fertilizer sources for maize and wheat. We applied the recommended rate of Glyphosate (N-(phosphonomethyl) glycine) for weed control in conventional maize and soybean and applied a fungicide (Pyraclostrobin + Fluxapyroxad) on high input soybean as well as an herbicide (thifensulfuron + tribenuron) and fungicide (Prothioconazole + Tebuconazole) on high input wheat. In organic rotations, we used the non-treated seed of a non-GM maize hybrid (isoline to the

conventional GM hybrid), a non-GM soybean variety, and the same wheat cultivar as in conventional wheat. We used composted organic manure (5-4-3, N-P-K analysis) for starter fertilizer and as an N source for maize and wheat. We used tine weeding or rotary hoeing plus 3 to 4 cultivations on maize and soybean, and we used an organic seed treatment (SaBrex that has different strains of Trichoderma, soil-dwelling fungi) in high input maize, soybean, and wheat. Please refer to our previous manuscript for more details on each crop [15]. Costs for the management inputs that differed between organic and conventional systems for the three crops are listed in Table S1. Production costs for organic maize and wheat will be somewhat inflated because of the use of composted chicken manure as the major N source (~13× higher cost/kg of N compared with liquid N in conventional maize and ammonium nitrate in wheat) instead of cheaper solid dairy manure. We used composted chicken manure in organic maize and wheat because of its known analyses of N-P-K and its ease in calibration and application with a fertilizer spreader. We wished to avoid the problems associated with the use of solid animal manure in previous studies, which did not accurately estimate the N content [4,6].

**Table 2.** Management inputs that differed between farming systems, including planting rate, seed treatment, cultivar, starter and N fertilizer practices, weed control practices, and disease control practices for maize, soybean, and wheat in conventional and organic farming systems with two management treatments (recommended and high input) at a Cornell research farm in central New York, USA in 2017 and 2018.

| Descriptor | Crop | | | | | |
|---|---|---|---|---|---|---|
| | Maize | | Soybean | | Wheat | |
| | Rec. | High | Rec. | High | Rec. | High |
| | **Conventional** | | | | | |
| Planting rate (seeds/ha) | 73,100 | 87,700 | 370,500 | 494,000 | 2,964,000 | 4,200,000 |
| Seed treatment | Fungicide/insecticide | | Fungicide/insecticide/inoculant | | Fungicide/insecticide | |
| Cultivar | GM hybrid | GM hybrid | GM variety | GM variety | Soft white (P24R46) | Soft white (P24R46) |
| Starter fertilizer (kg/ha) | 280 kg/ha (10-20-20) | | None | | 225 kg/ha (10-20-20) | |
| N fertilizer-side-dress (kg N/ha) | 56–135 kg [+] N/ha (liquid) | 112–180 kg [+] N/ha (liquid) | None | None | 80 kg N/ha (33-0-0) | 56 + 56 kg N/ha (33-0-0) |
| Herbicide application | Glyphosate | Glyphosate | Glyphosate | Glyphosate | None | Yes |
| Fungicide application | None | None | None | Yes | None | Yes |
| | **Organic** | | | | | |
| Planting rate (kernels/acre) | 73,100 | 87,700 | 370,500 | 494,000 | 2,964,000 | 4,200,000 |
| Seed treatment | None | SaBrex | None | SaBrex | None | SaBrex |
| Cultivar | Non-GM Isoline | Non-GM Isoline | Non-GM variety | Non-GM variety | Soft white (P24R46) | Soft white (P24R46) |
| Starter fertilizer | 350 kg/ha composted chicken manure (5-4-3) | | None | | 175 kg N/ha composted chicken manure (5-4-3) | |
| Pre-plant N fertilizer (kg N/ha) | 0–135 kg N/ha composted manure | 56–180 kg N/ha composted manure | None | None | 80 kg N/ha compostmanure | 56 + 56 kg N/ha compost manure |
| Tine weeding/rotary hoeing Cultivate | 1× 3× | | 1× 4× | | None None | |

[+] The lower rate was applied in the S-W/Cl-M-S rotation and the higher rate in the M-S-M-S rotation. Rec., recommended; GM, genetically modified.

Conventional maize prices received by New York farmers averaged $0.156/kg in 2015 and 2016, and $0.151/kg in 2017 [16]. Conventional soybean prices received by New York farmers averaged $0.345/kg in 2015 and 2016, $0.341 in 2017, and $0.316/kg in 2018. Conventional wheat prices received by New York farmers averaged $0.149/kg in 2016 and $0.174/kg in 2018. Organic/conventional price ratios averaged 2:1 for maize in 2017, 1:85 for soybean in 2017 and 2018, and 1:54 for wheat in 2018. Economic analyses focused on enterprise budget items that differed among the treatments, namely the value of production associated with yield differences as well as cost differences for inputs for maize, soybean, and wheat. Our selected variable inputs include: seed, fertilizer, and sprays/other

inputs (herbicide, fungicide, inoculant, and organic seed treatment); labor and machinery operating inputs (repairs and maintenance, fuels, and lubricants), excluding tillage, planting and harvesting tasks, except for hauling where hauling cost is a function of yield [17]. Cost of production values reported for fixed inputs exclude farm machinery ownership costs for tillage, planting and harvest, land charges, and values of management inputs. Variable and fixed costs for maize and soybean in 2017 and wheat for 2018 are in Tables S2–S4. We did not provide a table for 2018 soybean because the costs are only a few dollars above costs for 2017 soybean and the differences between soybean farming systems and management inputs were essentially the same as in 2017. Returns above variable and fixed input costs that do not differ between conventional and organic production under recommended and high input management were calculated for the three crops.

Previous crops (or fields), farming systems (conventional and organic), crop rotations (Cl-M-S-W; M-S-M-S; and S-W/Cl-M-S), and management inputs (recommended and high) were considered fixed and replications random for statistical analyses of yield, revenue, and returns for each crop within individual years, using the REML (Residual Maximum Likelihood) function in the MIXED model from the Statistical Analysis System (SAS version 9.4; SAS Institute Inc., Cary, NC, USA). We also used the same SAS procedures to statistically analyze the revenue and partial returns of the three rotations for the two-year transition period, the first two years after transition when organic crops are eligible for the organic premium, and the entire four-year study period. We could not statistically analyze costs because the same inputs were applied to the four replications so there was no random variability and thus no statistical analysis. Fields with different previous crops (2014 crops) had yield differences for maize and soybean in 2015 but did not have any main effects or interactions with farming systems, rotations, or management inputs in subsequent years or in the two-year or four-year economic analyses. Consequently, the data will be pooled across previous crops (the three contiguous fields) for the 2017 and 2018 years, and across the two-year and four-year data sets. Least square means of the main effects (farming system, crop rotations, and management inputs) were computed and means separations were performed on significant effects using Tukey's studentized range test (HSD), with statistical significance set at $p < 0.05$. Differences among least square means for farming system interactions were calculated also using the HSD test. Two-way and three-way interactions were detected for most variables so the interaction comparisons will be presented.

## 3. Results and Discussion

### 3.1. Individual Crops

When averaged across the 2015 and 2016 growing seasons, organically-managed maize compared with conventional maize had ~$785/ha lower returns above selected production costs because of ~$440/ha higher costs coupled with a 32% lower yield in 2015 and a 7% lower yield in 2016 [18]. In 2017, however, organic compared with conventional maize had greater returns, despite mostly higher production costs (Table S2), because organic compared with conventional maize yielded similarly (M-S-M-S rotation) or higher (S-W/Cl-M-S rotation) and the organic premium was in place. (Table 3). Returns, however, had a farming system × crop rotation interaction. Organic compared with conventional maize had $1007/ha higher returns in the M-S-M-S rotation but $2107/ha higher returns in the S-W/Cl-M-S rotation. (Table 3). The ~2× higher comparative organic returns in the S-W/Cl-M-S rotation compared to the M-S-M-S rotation are because of higher yields but also because of lower production costs, mostly because of much lower fertilizer costs (Table S2). Consequently, the interseeding of red clover into the preceding wheat crop was not only an effective N source for organic maize, as evidenced by the high organic maize yields in the S-W/Cl-M-S rotation, but also reduced the amount of expensive composted manure, as evidenced by lower production costs for organic maize in the S-W/Cl-M-S compared to the M-S-M-S rotation (Table S2). Returns above selected costs also had a crop rotation × input management interaction with conventional and organic maize showing a significant response to high input management in the M-S-M-S rotation (17.4% and 9.8%,

respectively) but no significant response in the S-W/Cl-M-S rotation (Table 3). The greater returns for organic compared with conventional maize in both rotations in 2017 are higher than reported by other researchers [3–7,11,12] because of the lack of the typical yield penalty for organic maize in the third year of our study.

**Table 3.** Yield, revenue, and returns above selected production costs for maize in 2017 in the maize-soybean-maize-soybean (M-S-M-S) crop rotation and soybean-wheat/clover-maize-soybean (S-W/Cl-M-S) crop rotation under conventional and organic farming systems at recommended and high input management at a Cornell University research farm in central New York, USA.

| Treatment | M-S-M-S | S-W/Cl-M-S |
|---|---|---|
| | Yield (kg/ha) | |
| **Conventional** | | |
| Recommended | 10,556 [+] b | 10,145 c |
| High Input | 12,547 a | 11,014 b |
| **Organic** | | |
| Recommended | 10,294 b | 11,301 b |
| High Input | 12,001 a | 12,952 a |
| SE [++] | 228 | |
| | Revenue ($/ha) | |
| **Conventional** | | |
| Recommended | 1600 [+] d | 1538 c |
| High Input | 1902 c | 1670 c |
| **Organic** | | |
| Recommended | 3277 b | 3597 b |
| High Input | 3821 a | 4123 a |
| SE [++] | 53 | |
| | Returns ($/ha) | |
| **Conventional** | | |
| Recommended | 910 [+] c | 902 b |
| High Input | 1068 c | 899 b |
| **Organic** | | |
| Recommended | 1902 b | 2983 a |
| High Input | 2089 a | 3033 a |
| SE [++] | 53 | |

[+] Treatment means within the same column followed by the same letter are not significantly different according to Tukey's studentized range test (HSD) at $p < 0.05$ level. [++] SE is the standard error of the mean for the farming system × crop rotation × input management interaction.

Organically-managed soybean and conventional soybean had mostly similar yields, production costs, and returns in 2015 and 2016 [15]. Returns above the selected production costs had a farming systems effect with no interactions in 2017 and 2018. Organic compared with conventional soybean had ~$800/ha to ~$900 higher returns in 2017 and 2018, despite ~9 to ~14% lower yields (Tables 4 and 5) and similarly higher production costs (Table S3), because of the organic price premium. As with maize, returns of organic compared with conventional soybean in the third and fourth years of our study greatly exceeded those reported in the USDA survey [3] and the Maryland, USA study [12]. The USDA report and Maryland study, however, reported 32% to 40% lower yields for organic compared with conventional soybean. Our much higher returns in this study are obviously because of the more competitive yields of organic soybean. Perhaps, pest issues, especially weed control challenges, will arise in future years in organic soybean, which would increase the yield penalty 9%–14% to values reported in other studies.

**Table 4.** Yield, revenue, and returns above selected production costs for soybean and wheat in 2017 and 2018 in the clover-maize-soybean-wheat rotation under conventional and organic farming systems with recommended and high input management at a Cornell University research farm in central New York, USA.

| Treatment | Yield (kg/ha) | Revenue | Returns ($/ha) |
|---|---|---|---|
| **2017 Soybean** | | | |
| **Conventional** | | | |
| Recommended | 3962 [+] a | 1353 b | 1080 b |
| High Input | 4085 a | 1397 b | 989 b |
| **Organic** | | | |
| Recommended | 3564 b | 2252 a | 1932 a |
| High Input | 3660 b | 2314 a | 1921 a |
| SE[++] | 61 | 31 | 31 |
| **2018 Wheat** | | | |
| **Conventional** | | | |
| Recommended | 5360 [+] b | 935 c | 536 b |
| High Input | 5320 b | 927 c | 311 c |
| **Organic** | | | |
| Recommended | 5346 b | 1438 b | 639 a |
| High Input | 5754 a | 1547 a | 459 b |
| SE [++] | 126 | 27 | 27 |

[+] Treatment means within the same column followed by the same letter are not significantly different according to Tukey's studentized range test (HSD) at $p < 0.05$ level. [++] SE is the standard error of the mean for the farming system × crop rotation × input management interaction.

**Table 5.** Yield, revenue, and returns above selected production costs for soybean in 2018 in the maize-soybean-maize-soybean (M-S-M-S) and soybean-wheat/clover-maize-soybean (S-W/Cl-M-S) crop rotations under conventional and organic farming systems at recommended and high input management at a Cornell University research farm in central New York, USA.

| Treatment | M-S-M-S | S-W/CL-M-S |
|---|---|---|
| **Yield (kg/ha)** | | |
| **Conventional** | | |
| Recommended | 3919 [+] a | 3793 b |
| High Input | 4100 a | 4159 a |
| **Organic** | | |
| Recommended | 3582 b | 3597 b |
| High Input | 3486 b | 3655 b |
| SE [++] | 109 | |
| **Revenue ($/ha)** | | |
| **Conventional** | | |
| Recommended | 1238 b | 1199 c |
| High Input | 1296 b | 1314 b |
| **Organic** | | |
| Recommended | 2094 a | 2103 a |
| High Input | 2038 a | 2136 a |
| SE [++] | 61 | |
| **Returns ($/ha)** | | |
| **Conventional** | | |
| Recommended | 958 [+] b | 919 b |
| High Input | 880 b | 897 b |
| **Organic** | | |
| Recommended | 1757 a | 1766 a |
| High Input | 1628 a | 1725 a |
| SE [++] | 61 | |

[+] Treatment means within the same column followed by the same letter are not significantly different according to Tukey's studentized range test (HSD) at $p < 0.05$ level. [++] SE is the standard error of the mean for the farming system × crop rotation × input management interaction.

Organically-managed wheat compared with conventional wheat had lower (11.5%) to similar yields, ~$400 to ~$600 higher production costs, and ~$550 to ~$660 lower returns in recommended and high input management, respectively, in 2016 [15]. Returns above selected production costs had a farming system and input management effect but no interactions in 2018. Organic compared with conventional wheat had ~$125/ha greater returns, despite ~$440/ha greater total selected production costs (Table S4) and similar to modestly higher yields, because of the organic price premium (Table 5). The wheat returns in 2018, however, were lower than the $245/ha greater return for organic compared to no-till conventional wheat across seven years of data in the study at Maryland, USA [12]. The lower returns in our study compared with the Maryland study, despite our high organic wheat yields, is associated with the greater costs for composted chicken manure in our study compared to the composted chicken litter used in the Maryland study. The $125/ha greater returns for organic compared with conventional wheat in our study, however, is greater than the $9/ha advantage for organic wheat in the USDA survey study [3]. Despite the 7.6% higher organic wheat yield with high inputs compared with recommended inputs, the recommended input treatment had $180/ha higher returns (Table 5). Obviously, the greater revenue for organic wheat with high input management did not offset the increased costs for the composted chicken manure.

*3.2. Crop Rotations*

The organic compared with the conventional farming system had similar to $692/ha lower returns above selected production costs in the Cl-M-S-W rotation, $1134/ha to $1363/ha lower returns in the M-S-M-S rotation, and $548/ha to $662/ha lower returns in the S-W/Cl-M-S rotation with recommended and high inputs, respectively, during the transition phase in 2015 and 2016 [15] (Table 6). We also reported that most conventional growers would not plant a green manure crop as part of their rotation so comparisons should be made between the organic Cl-M-S-W rotation (organic growers sometimes use a green manure crop in the first year of the transition) with the conventional M-S-M-S rotation. In this comparison, the organic compared with the conventional farming system had $1127/ha lower returns with recommended inputs and $1786/ha lower returns with high inputs. Consequently, we suggested that the best strategy for a transitioning grower to an organic farming system in our environment may be to select soybean as the entry crop (instead of using a green manure crop) during the first year of transition followed by wheat/red clover because the comparative loss in returns in this rotation would be the least [15].

Returns above selected production costs had a farming system × crop rotation interaction in 2017 and 2018, the first two years when the organic price premium was in place. The farming system × crop rotation interaction was because of much greater comparative differences in returns between farming systems across the three crop rotations (Table 7). When averaged across input treatments, organic compared with the conventional system had $1018/ha higher returns in the Cl-M-S-W rotation, $1782/ha higher returns in the M-S-M-S rotation, and $2961/ha higher returns in the S-W/Cl-M-S rotation. The greater organic compared with the conventional returns were much higher than that reported in other studies comparing these rotations under conventional and organic farming systems [3–7,11,12]. When averaged across input treatments, the organic S-W/Cl-M-S rotation had $1080/ha higher returns than the organic M-S-M-S rotation and $2293/ha higher returns than the organic Cl-M-S-W rotation. As with the transition phase, the S-W/Cl-M-S rotation provided the greatest returns in the two years after the transition phase. The organic compared with the conventional farming system had much greater returns because of much greater revenues in 2017 and 2018, associated with similar to greater maize and wheat yields or slightly lower soybean yields, coupled with the organic premiums (Table 7). The mostly higher production costs for organic compared with the conventional system did not offset the higher revenue, resulting in much higher returns for the organic crop rotations (Table 7).

**Table 6.** Total selected production costs, total revenue, and total returns above selected production costs of the red clover-maize-soybean-wheat (Cl-M-S-W), maize-soybean-maize-soybean (M-S-M-S), and the soybean-wheat/red clover-maize-soybean (Cl-M-S-W) crop rotations during the transition period (2015 and 2016) in conventional and organic farming systems with recommended and high input management at a Cornell University research farm in central New York, USA.

| | Sequence during Transition (2015–2016) | | |
|---|---|---|---|
| Treatment | Cl-M-S-W | M-S-M-S | S-W/Cl-M-S |
| | Total Costs ($/ha) | | |
| **Conventional** | | | |
| Recommended | 741 | 958 | 605 |
| High Input | 909 | 1211 | 956 |
| **Organic** | | | |
| Recommended | 666 | 1556 | 1035 |
| High Input | 1503 | 2077 | 1505 |
| | Total Revenue ($/ha) | | |
| **Conventional** | | | |
| Recommended | 1116[+] a | 2526 a | 1647 a |
| High Input | 1214 a | 2610 a | 1679 a |
| **Organic** | | | |
| Recommended | 1107 a | 1990 b | 1530 a |
| High Input | 1116 a | 2041 b | 1567 a |
| SE[++] | | 54 | |
| | Total Returns ($/ha) | | |
| **Conventional** | | | |
| Recommended | 375[+] a | 1568 a | 1043 a |
| High Input | 305 a | 1399 a | 724 b |
| **Organic** | | | |
| Recommended | 441 a | 434 b | 495 c |
| High Input | −387 c | −36 c | 62 d |
| SE[++] | | 65 | |

[+] Treatment means within the same column followed by the same letter are not significantly different according to Tukey's studentized range test (HSD) at $p < 0.05$ level. [++] SE is the standard error of the mean for the farming system × crop rotation × input management interaction.

Returns above selected production costs of the four-year crop rotations also had a farming system × crop rotation interaction because of much greater comparative differences in returns between farming systems across the three crop rotations (Table 8). When averaged across input treatments, the organic compared with the conventional system had $673/ha higher returns in the Cl-M-S-W rotation, $497/ha higher returns in the M-S-M-S rotation, but $2355/ha higher returns in the S-W/Cl-M-S rotation. Overall, the organic S-W/Cl-M-S rotation, when averaged across input treatments, had $1158/ha higher returns than the organic M-S-M-S rotation and $2545/ha higher returns than the organic Cl-M-S-W rotation. Obviously, planting a green manure crop (red clover) as the entry crop to the transition phase, resulting in the subsequent Cl-M-S-W/Cl rotation, was the least profitable organic rotation in the short-term (four-year period of this study). Despite the lower returns for organic wheat compared with organic soybean or organic maize, the inclusion of wheat in the organic rotation was far more profitable than the M-S-M-S rotation. In contrast, the M-S-M-S rotation was most profitable for the conventional cropping system. Other researchers also have reported much greater returns for organic systems with a more extended rotation involving more crops [15]. In the Maryland, USA study [12], however, the three-crop organic rotation (variation on the S-W/Cl-M-S rotation) had similar returns to the two-crop organic rotation (M-S rotation). In the Maryland study, the inclusion of three years of alfalfa to a

rotation with maize, soybean, and wheat, resulting in a six-year rotation, was far more profitable than the two-year or three-year organic rotations.

**Table 7.** Total selected production costs, total revenue, and total returns above selected production costs of the red clover-maize-soybean-wheat (Cl-M-S-W), maize-soybean-maize-soybean (M-S-M-S), and the soybean-wheat/red clover-maize-soybean (Cl-M-S-W) crop rotations after the transition period (2017 and 2018) in conventional and organic farming systems with recommended and high input management at a Cornell University research farm in central New York, USA.

| | Sequence after Transition (2017–2018) | | |
|---|---|---|---|
| **Treatment** | **Cl-M-S-W** | **M-S-M-S** | **S-W/CL-M-S** |
| | **Total Costs ($/ha)** | | |
| **Conventional** | | | |
| Recommended | 672 | 970 | 916 |
| High Input | 1024 | 1250 | 1188 |
| **Organic** | | | |
| Recommended | 1119 | 1708 | 937 |
| High Input | 1480 | 2142 | 1483 |
| | **Total Revenue ($/ha)** | | |
| **Conventional** | | | |
| Recommended | 2288 [+] c | 2838 d | 2737 d |
| High Input | 2324 c | 3198 c | 2984 c |
| **Organic** | | | |
| Recommended | 3690 b | 5371 b | 5700 b |
| High Input | 3861 a | 5859 a | 6259 a |
| SE[++] | | 49 | |
| | **Total Returns ($/ha)** | | |
| **Conventional** | | | |
| Recommended | 1616 [+] c | 1868 b | 1821 b |
| High Input | 1300 d | 1948 b | 1797 b |
| **Organic** | | | |
| Recommended | 2571 a | 3663 a | 4763 a |
| High Input | 2380 b | 3717 a | 4776 a |
| SE [++] | | 49 | |

[+] Treatment means within the same column followed by the same letter are not significantly different according to Tukey's studentized range test (HSD) at $p < 0.05$ level. [++] SE is the standard error of the mean for the farming system × crop rotation × input management interaction.

Returns above selected production costs in the four-year rotations also had a crop rotation × input management interaction because of much greater comparative differences in returns between input treatments across the three crop rotations. When averaged across farming systems, recommended compared to high input management had 11.2% higher returns in the S-W/Cl-M-S rotation, 7.2% higher returns in the M-S-M-S rotation, but 37.6% higher returns in the Cl-M-S-W rotation. Despite the crop rotation × management input interaction, it is clear that organic and conventional farming systems, regardless of rotation, did not respond to high input management. Many organic growers use higher than recommended seeding rates with the goal of improved weed control. In our study, we saw statistically fewer weeds with high input management in maize, soybean, and wheat but differences were so small that it had no effect on crop yield in this environment [15,18,19]. The results of this study indicate that the use of higher seeding and N rates is not justified in the first two transition years and the first two years after organic certification of maize, soybean, and wheat management under the environmental conditions of this study.

**Table 8.** Total selected production costs, total revenue, and total returns above selected production costs of the red clover-maize-soybean-wheat (Cl-M-S-W), maize-soybean-maize-soybean (M-S-M-S), and the soybean-wheat/red clover-maize-soybean (Cl-M-S-W) crop rotations during the four-year period (2015 through 2018) in conventional and organic farming systems with recommended and high input management at a Cornell University research farm in central New York, USA.

| | Four-Year Sequence (2015–2018) | | |
|---|---|---|---|
| Treatment | CL-M-S-W | M-S-M-S | S-W/CL-M-S |
| | Total Costs ($/ha) | | |
| **Conventional** | | | |
| Recommended | 1413 | 1928[+] | 1521 |
| High Input | 1932 | 2460 | 2144 |
| **Organic** | | | |
| Recommended | 1785 | 3264 | 1972 |
| High Input | 2983 | 4219 | 2988 |
| | Total Revenue ($/ha) | | |
| **Conventional** | | | |
| Recommended | 3405 b | 5364 d | 4384 c |
| High Input | 3538 b | 5808 c | 4663 c |
| **Organic** | | | |
| Recommended | 4797 a | 7361 b | 7230 b |
| High Input | 4977 a | 7900 a | 7826 a |
| SE[++] | | 108 | |
| | Total Returns ($/ha) | | |
| **Conventional** | | | |
| Recommended | 1992 b | 3436 bc | 2863 c |
| High Input | 1605 c | 3347 c | 2520 d |
| **Organic** | | | |
| Recommended | 3012 a | 4097 a | 5257 a |
| High Input | 1993 b | 3681 b | 4838 b |
| SE[++] | | 107 | |

[+] Treatment means within the same column followed by the same letter are not significantly different according to Tukey's studentized range test (HSD) at $p < 0.05$ level. [++] SE is the standard error of the mean for the farming system × crop rotation × input management interaction.

## 4. Conclusions

Our results indicate that field crop producers who transition to organic maize, soybean, and wheat production can be more profitable than conventional field crop producers after four years under the environmental conditions of this study, if they can survive the cash-flow challenge during the transition. To minimize the cash-flow challenge, transitioning growers should not apply prohibited inputs for use in organic systems after late spring in their last conventional crop so the 36-month transition period can be accomplished in two growing seasons. Results of this study indicate that transitioning growers should not use a green manure crop in the first year of transition but rather plant soybean in the first year. Soybean does not require N fertilizer, a major constraint to organic maize and wheat production. Soybean is also competitive with weeds, especially in conjunction with aggressive cultivation. Although the previous conventional crop before transition did not affect yield, revenue or returns above selected production costs in this study, we recommend that growers time their transition to fields where soybean is the intended crop.

Results from the study indicate that field crop producers in the Northeast, USA should also include winter wheat as the second crop in the transition after soybean, although it was the least

profitable organic crop. Growers should also frost-seed red clover into standing wheat in early spring, a typical practice for many conventional wheat growers in the Northeast, USA.

Results of the study indicate that field crop producers in the Northeast, USA, who transition to an organic cropping system, should plant maize in the third year, the first year when crops are eligible for the organic premium. Organic maize typically has a higher organic premium when compared with organic soybean and organic wheat. Maize should be planted following wheat, interseeded with red clover, which provides considerable slow-release N to the subsequent maize crop. In addition, the wheat/red clover crops can disrupt weed cycles, as evidenced by the much lower weed densities in organic maize in the S-W/Cl-M-S compared with the M-S-M-S rotation in this study [18]. In the fourth year of the study, field crop producers in New York who have adopted an organic farming system should begin the S-W/Cl-M rotation again by planting soybean.

Results of this study indicate that growers should use current recommended inputs for conventional crops and not use elevated seeding rates to improve weed control in the three crops or use higher N rates to provide more available soil N to maize and wheat. Although the organic compared to the conventional farming system was more profitable in this study, we recognize that commodity prices, farm size, political beliefs, and other factors influence a grower's decision on whether to transition to an organic cropping system [14]. Also, we recognize that growing conditions at our location were unique to this study, so results may differ in different years and locations in the Northeast, USA.

**Supplementary Materials:** The following are available online at http://www.mdpi.com/2073-4395/9/8/424/s1, Table S1. Costs of variable inputs, including seed, hopper seed treatments, (inoculant for conventional soybean and SaBrex for organic crops), starter fertilizer, N fertilizer, herbicide, and fungicide in conventional and/or organic soybean in 2017 and 2018, maize in 2017, and wheat in 2018; Table S2. Income, selected production costs, and returns above selected production costs for conventional maize with recommended management (M1) and high input management (M2) and organic maize with recommended management (M3) and high input management (M4) in a soybean-wheat/red clover-maize-soybean rotation, and conventional maize with recommended management (M5) and high input management (M6) and organic maize with recommended management (M7) and high input management (M8) in a maize-soybean-maize-soybean rotation at a Cornell University research farm in central New York, USA in 2017; Table S3. Income, selected production costs, and returns above selected production costs for conventional soybean with recommended management (S1) and high input management (S2); and organic soybean with recommended management (S3) and high input management (S4) at a Cornell University research farm in central New York, USA in 2017; Table S4. Income, selected production costs, and returns above selected production costs for conventional wheat with recommended management (W1) and high input management (W2); and organic soybean with recommended management (W3) and high input management (W4) at a Cornell University research farm in central New York, USA in 2018.

**Author Contributions:** W.C. designed and conducted the experiment, and wrote the initial draft of the manuscript. J.J.H. conducted the economic analyses for the study. J.C. conducted the statistical analyses and reviewed the manuscript. M.S. is the PI of the project and reviewed the manuscript.

**Acknowledgments:** This research was partially funded by the U.S. Department of Agriculture Cooperative State Research, Education, and Extension Service through New York Hatch Project 1257322. Pioneer Hi-Bred supplied all the seed for the study.

**Conflicts of Interest:** The authors have no conflicts of interest.

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
