# Peer review of "Economic Responses of Maize, Soybean, and Wheat in Three Rotations under Conventional and Organic Systems"

_agronomy, doi:10.3390/agronomy9080424_

Round 1

Reviewer 1 Report

Review

Economic responses of maize, soybean, and wheat in three rotations under conventional and organic cropping systems

Topic:

Cropping system experiment is crucial to address the several challenges of agriculture. Comparing organic agriculture (OA) and conventional agriculture on rentability is important. The general topic is thus very interesting.

However, the specific topic of the article remains unclear to me. It is supposed to emphasize what is the best entry crop (or rotation) when transitioning to organic agriculture but the article mixes a lot of different results (annual vs pluriannual) and the reader loses the main message of the article. To my mind, it seems that this topic was already presented in previous articles of the same author, published in the same journal (“Agronomic and Economic Performance of Maize, Soybean, and Wheat in Different Rotations during the Transition to an Organic Cropping System” and “Agronomic Comparisons of Conventional and Organic Maize during the Transition to an Organic Cropping System”). In consequence, I am not sure of the originality of the results presented here.

General comment:

This paper has repetitions that can be avoided. Globally, I found the paper confusing and not easily readable. A lot of concept definitions are unclear, as well as the material and methods that need to be more synthetic, eventually adding a supplementary material. It is hard to understand when and in which rotation the different crops were present.

As examples concerning the blurring definitions of concepts, at least two major problems appear:

-          “organic” and “conventional” ARE NOT cropping systems and cannot be called like that. Please refer to the (huge) existing literature on that topic and definition (e.g. https://link.springer.com/chapter/10.1007/978-1-4020-8709-7_7)

-          2015 and 2016 crops are sometimes presented as organic while they are under conversion! This is a big problem in the article because it is known that fertility from the previous years remains the firsts years of conversion. It’s even the reason why the conversion period is for.

The bibliography presented in introduction is too poor and focused on too few authors respect to the important literature present on the economic evaluation/comparison of organic and conventional cropping systems. Why focusing only on three/four states in the US?

By the way, the literature presented is too much detailed and I lost the synthesis and general idea the authors want to prove with these previous studies. The introduction lacks in presenting the main issues that the article is supposed to answer (L118-122). The synthesis is lacking. Particularly, the objective “b” is not presented at all in the introduction!!!!! Moreover, the justification of the high input is not clear. What it is not presented in the objectives of the study? What is the hypothesis beneath that?

In material and methods, a diagram representing the different cultures for the different rotations across the years and plots would be helpful.

The results of the individual crops are superfluous and too much detailed respect to the “message” of the article. Globally, I had problems to understand the complementarity between the study of the individuals crops and the crop rotation. The individual crop study is very long and not informative enough.

In the all the paper, any standard deviation is presented which is however necessary to understand the variability of the results obtained, especially when post-hoc tests are realized. In general, tables are too poorly described.

Finally, I found the global conclusions – firstly the choice of soybean as the first crop of the rotation –  of the article too poorly supported by the results or maybe too much lost in others – less important – results.

Specific comments:

L35-37: Why these rotations? How much acres? Proportion?

L38: about the 36-month transition => it exists because fertility from the conventional agriculture are still present. This has to be presented.

L59-68: the results of the rotation are more pertinent to be presented

L98-100: data is too old because of the very high variability of the prices

L145: so not all the crops of a rotation were not present every year?

L171: What is “Sabrex”?

L172-179: not clear – rely more on the following table

L183: why going up to 86500? What is the justification?

Table 1: lack the name of the molecules different that glyphosate. Doses are also important. Supplementary material?

L197-L202: repetition with above. Too long on the cropping practices.

L213: why drill presented here and not elsewhere?

L221: “insecticide/fungicide” => which molecules?

L222: 175kg/ha?? 170kg/ha in the table

L231: both years??? Wheat was not only present in 2018 ?

L245-246: not material and methods. Has to be supported by literature in introduction

Table 2: superfluous

L266-268: explain why some costs are included and not others

L282: cropping system definition absolutely need to be changed

L295-298: sentence could be rewritten more simply

Table 3: add the standard deviations

L314: “4-time mechanical weeding” => how are they decided? 4-time is very much!! It has a huge impact on the evaluation of the rotations.

Table 4:                M1,M2,…M8 are not presented in material&methods

                                Too long table, too much data not presented and not useful. Be more synthetic and present the rest in a supplementary material

L362-365: why presenting weed data? Data collection not presented in material and methods

L371-373: “Wider row…soybean production” => contradictory with the previous sentence

L396-398: “A Minnesota USA…” => interest of presenting this data

Table 7: why only 4 modalities and not 8 as in maize?

Table 7 and Table 8 are the same!!! Decimals not needed

L415: Why presenting the rotation if rotation has no effect

L428: crops in 2016 are not organic!!!! It remains previous fertility

L431-435: not clear

L440: “still do not use herbicides” => source??

L442-445: Why comparing high-conventional vs high-OA? What is the hypothesis beneath this?

L447: use of chicken manure is standard?

L485-487: “We also […] greatly.” => contradictory respect to the previous sentence.

Table 10: standard deviations lacking

L504-506: tests not presented in table 11 (only between rotations, not organic vs conventional)!!

L519-521: Results presented here are not significative

L533: S-W/Cl-M-S has good partial returns only because 2015 and/or 2016 were favourable for the crops?

L542: + introduction of cover-crops ?

L544-549: too much centred on an example

Table 12: add standard deviations

L561: conventional neither responded to high input

L562-563: “Many organic growers […] weed control” => what is the data? Why not presented in context?

L566: “first 4 years” => it is in fact transition 2 years conversion + 2 years OA

L578-580: Not enough supported by the results presented. How to manage the introduction of soybean in first year of conversion to OA at the level of an entire farm?

L582-586: it is very surprising, that crops or rotations can be suggested for an entire region. I doubt that the techniques presented here can be transposed identically on an entire region. See literature on rule-based cropping systems.

Author Response

Response to Reviewer #1

Thank-you for your very thorough review and I apologize for your frustrating experience reviewing it. Admittedly, it was too long. But it is a 4-year study with 3 fields, 2 farming systems, 3 crop rotations, and 2 management input so there is going to be a lot of data. And I just hate to gloss over thing by saying “higher” or “lower” and not providing specific data. I am afraid that I am too data-driven and it shows up in my writing and speaking for that matter. I have shortened the manuscript significantly by reducing the Introduction by 27% (from 1277 down to 887 words-there I go again providing data!), M&M section by 43% (2206 words down to 1257 words), the discussion on the individual crops by 62% (1863 to 710 words), and the discussion on the crop rotations by 18% (1087 to 877 words).

 “To my mind, it seems that this topic was already presented in previous articles of the same author, published in the same journal (“Agronomic and Economic Performance of Maize, Soybean, and Wheat in Different Rotations during the Transition to an Organic Cropping System” and “Agronomic Comparisons of Conventional and Organic Maize during the Transition to an Organic Cropping System”). In consequence, I am not sure of the originality of the results presented here”. I believe that this is a totally unfair assessment-all data presented (except for Table 10, which I have kept in the manuscript as the new Table 6 because it is integral to the new Table 8 showing the 4-year data set) is new data from the 3rd and 4th years of the study as well as the economic data from the 4-years. I disagree strongly with this assessment.

General Comments

 “This paper has repetitions that can be avoided. Globally, I found the paper confusing and not easily readable. A lot of concept definitions are unclear, as well as the material and methods that need to be more synthetic, eventually adding a supplementary material. It is hard to understand when and in which rotation the different crops were present”. Hopefully, shortening the different sections will alleviate some of the confusion and lack of readability. In addition, I have added a Table (Table 1 in the revision) that clearly shows the different crops in their respective rotations across years. I apologize for not providing that table in the first submission.

“organic” and “conventional” ARE NOT cropping systems and cannot be called like that. Please refer to the (huge) existing literature on that topic and definition (e.g. https://link.springer.com/chapter/10.1007/978-1-4020-8709-7_7). Wow-I studied agronomy in the 1970s so must have forgotten these definitions. In addition, I have been a University agronomist for 40 years and never ran into these definitions. Evidently, I am not the only one (if you look at citations 5,6, and 13 and read the titles you will see that they are referred to as “organic cropping systems”-two in Agronomy Journal and one in the Proceedings of the National Academy of Science). But you are correct, as indicated in the link that you provided. But I am still not sure how to refer to them-I have used “farming systems” when discussing it in statistical terms and otherwise have just used “organic systems” as White et al. (2019) have done in their recent publication in Agronomy Journal.

“2015 and 2016 crops are sometimes presented as organic while they are under conversion! This is a big problem in the article because it is known that fertility from the previous years remains the firsts years of conversion. It’s even the reason why the conversion period is for”. Excellent point-I have amended and refer to those crops as “organically-managed”.

“The bibliography presented in introduction is too poor and focused on too few authors respect to the important literature present on the economic evaluation/comparison of organic and conventional cropping systems. Why focusing only on three/four states in the US??????. I focused on these studies because those are the studies that matched the crop rotations in my study (maize-soybean or maize-soybean-wheat/red clover). I don’t think it would be appropriate to cite all organic and conventional comparisons (why would I discuss a comparison of rice-based rotations or cotton-based rotations or rotations with different vegetable crops-these rotations are not relevant to my study). If I am writing a manuscript on the economics of corn seeding rates, I am not going to include literature on the economics of soybean or wheat seeding rates.

“By the way, the literature presented is too much detailed and I lost the synthesis and general idea the authors want to prove with these previous studies. The introduction lacks in presenting the main issues that the article is supposed to answer (L118-122). The synthesis is lacking. Particularly, the objective “b” is not presented at all in the introduction!!!!! Moreover, the justification of the high input is not clear. What it is not presented in the objectives of the study? What is the hypothesis beneath that?”

As mentioned above, I have shorted the Introduction by 27% and have combined a couple of paragraphs and eliminated another paragraph. I have rearranged the Introduction to highlight crop rotations and not individual crops, which I hope allays some of your concerns. The reason that there is no literature on objective “b” is that there are no studies comparing the maize-soybean and maize-soybean-wheat/red clover rotations in organic systems with high input management. I will have gladly added those studies if I had found them.

“In material and methods, a diagram representing the different cultures for the different rotations across the years and plots would be helpful”. Excellent suggestion-I have added Table 1.

The results of the individual crops are superfluous and too much detailed respect to the “message” of the article. Globally, I had problems to understand the complementarity between the study of the individuals crops and the crop rotation. The individual crop study is very long and not informative enough. I disagree that showing the results of the individual crops are superfluous-you need the individual crop data to understand the crop rotation data. I do agree that the Individual crops section was too long and I have shortened it by 62%. But I have left the individual crop data in the manuscript similar to the crop data provided in other studies that I have cited in my manuscript.

In the all the paper, any standard deviation is presented which is however necessary to understand the variability of the results obtained, especially when post-hoc tests are realized. In general, tables are too poorly described. Excellent point. I should have provided a measure of variability (standard deviation or standard error) in the tables-I have rectified and have provided standard errors for yield, revenue, and returns data (can’t provide one for the costs because the costs are the same across all 4 replications so no random variability to measure on costs). But because of the complexity of the study, I opted to provide only the standard errors of the farming systems x crop rotation x input management interactions (and not the standard errors for the main effects of the farming systems, the crop rotation or the input management and all their 2-way interactions).

Finally, I found the global conclusions – firstly the choice of soybean as the first crop of the rotation –  of the article too poorly supported by the results or maybe too much lost in others – less important – results. Probably got lost in the less important results but if you just compared the returns of the three organic rotations in the 2-year sequence during the transition, the 2 years after the transition,  and the 4-year sequence, the soybean-wheat/red clover-maize-soybean organic rotation is by far the most economical. I have tried to emphasize that more in this revision.

Specific Comments

L35-37: Why these rotations? How much acres? Proportion? Because these are the major field crops grown by cash crop producers in the Northeast USA-I do not think that I need to justify why I selected the crops that are grown in my region. I just looked at some previous economic articles on crop rotations and nobody justified why they selected the crops that are grown in their region and why they are studying them. Plus, you correctly pointed out that the original submission of the manuscript was way too long with too much data so I really don’t want to add more data on something that is really not necessary to justify.

L38: about the 36-month transition => it exists because fertility from the conventional agriculture are still present. This has to be presented. Excellent point-Have added a sentence stating this.

L59-68: the results of the rotation are more pertinent to be presented. Excellent suggestion-I have rearranged the paragraph to emphasize the rotations.

L98-100: data is too old because of the very high variability of the prices. Excellent point-I have eliminated most of this paragraph.

L145: so not all the crops of a rotation were not present every year? Correct-see Table 1 in the revision.

L171: What is “Sabrex”? Good question-I should have explained it in the original submission. It’s an organic seed treatment consisting of different Trichoderma strains or soil fungi strains-I have added a sentence explaining it in the M&M section

L172-179: not clear – rely more on the following table. I have rewritten to hopefully clarify the different inputs to organic and conventional farming systems.

L183: why going up to 86500? What is the justification? This is the typical seeding rate for many mainstream growers-~12,000 more than what I recommend. My peers at other Land Grant Universities in the Northern latitude of USA also recommend this-likewise, organic grower in my region are led  to believe that higher seeding rates are required for better weed control. I have conducted previously published studies comparing these two seeding rates (most recently, “Lack of Hybrid, Seeding and Nitrogen Rate Interactions for Corn Growth and Yield. Agronomy Journal, 104 (4), 945-952”) so I don’t think that I really need to justify the rate for this article. The M&M section was already too long and unwieldly, as you correctly pointed out, so I don’t need to add more verbiage that is really not necessary

Table 1: lack the name of the molecules different that glyphosate. Doses are also important. Supplementary material? It would be too difficult to fit into Table 1. I have mentioned it in the text now and have added recommended rate-the rate depends upon the formulation and the formulation changed over the years so don’t want to add extra verbiage and data when I am trying to reduce the amount of verbiage and data in the text. And I don’t want to add the specific names of the seed treatment chemicals-the seed treatments had three different fungicides that changed from year to year and from crop to crop and one insecticide, which was different for each crop and changed across years. It would take up a full paragraph just to state the chemical names of all the different seed treatment combinations and you have asked me below to shorten the text on cropping practices. I am looking to simplify the writing in this revision and adding all the chemical names of the seed treatments across years and crops would make the M&M section far more detailed and difficult to follow.

L197-L202: repetition with above. Too long on the cropping practices. Have shortened significantly.

L213: why drill presented here and not elsewhere? We use a grain drill to plant wheat and a maize planter to plant maize and soybean. The two are very different pieces of equipment that operate on very different principles.  This is the terminology that we use as agronomists to describe the different planting equipment in the USA.

L221: “insecticide/fungicide” => which molecules? Again-they changed across years and they are different for each crop. The list is far too long to include.  The chemicals used are specific to Pioneer-Hi-Bred cultivars-it is pre-treated seed by the company so it is not something that growers or we added to the seed.

L222: 175kg/ha?? 170kg/ha in the table. Sorry about that-it was 175.

L231: both years??? Wheat was not only present in 2018 ? See the new Table 1-wheat was harvested in 2016 and 2018.

L245-246: not material and methods. Has to be supported by literature in introduction-I have deleted this sentence.

Table 2: superfluous- I have moved it to the supplementary file as Table S1.

L266-268: explain why some costs are included and not others. That is what a partial budget analyses does. I have copied the definition of partial budgeting as defined by the Iowa State University Agricultural Economics Department.

Partial Budgeting: A Tool to Analyze Farm Business Changes: Business owners must often make decisions about changes they are either contemplating making or that have to be made. Many of the decisions are incremental, such as adding land, expanding or reducing an enterprise or changing how an enterprise is managed. The partial budget is a useful tool for farm managers when these situations arise. A partial budget helps farm owners/managers evaluate the financial effect of incremental changes.

“A partial budget only includes resources that will be changed. It does not consider the resources in the business that are left unchanged. Only the change under consideration is evaluated for its ability to increase or decrease income in the farm business”.

And I believe that the paragraph that I wrote on this in the M&M section explains this fully.

L282: cropping system definition absolutely need to be changed. Have changed it to “farming system”-if you told me what to change it to, I would have changed it to your definition.

L295-298: sentence could be rewritten more simply-have deleted.

Table 3: add the standard deviations-excellent suggestion-I have added the standard errors of the farming system x crop rotation x input management interaction to complement the data presentation of all the tables with yield, revenue, and returns (can’t add for costs because costs are the same for all 4 replications so no random error for costs).

L314: “4-time mechanical weeding” => how are they decided? 4-time is very much!! It has a huge impact on the evaluation of the rotations.-I don’t think 4x for maize and 5x soybean are too many. -we start with rotary hoeing (late May) and then follow with a very close cultivation to the row at the V3 stage or so (~June 10)-then 2 or 3 inter-row cultivations are routine for our region where summer annual and perennial weeds are prolific (V5-6 and again at the V7-8 stage in late June and early July). The critical weed-free period in maize is up to the V14 stage (~July 15) so we try to keep it as clean as possible up to the V14 stage. The critical weed-free period in soybean is up to the R4-R5 stage (~August 10) in soybean so we do another cultivation maybe around July 20-25.

Table 4: M1,M2,…M8 are not presented in material&methods. All the treatments are clearly marked in the table caption-they are coded as M1. M2…….M8 so they can fit into the tables-I can’t have the full treatment name in the header so have coded them and have explained the code in the table caption. Plus, I have moved this table to the supplementary file as Table S2.

Too long table, too much data not presented and not useful. Be more synthetic and present the rest in a supplementary material. I think that it is useful so would prefer that the readers see just what the costs are-the costs are critical to the crop rotation tables with costs, returns, and revenue. So I have retained all the costs but moved this table to the supplementary file, as suggested by you, and it is now Table S3.

L362-365: why presenting weed data? Data collection not presented in material and methods-good point-I have deleted.

L371-373: “Wider row…soybean production” => contradictory with the previous sentence-too confusing so I have deleted.

L396-398: “A Minnesota USA…” => interest of presenting this data-I have deleted.

Table 7: why only 4 modalities and not 8 as in maize? Refer to the new Table 1-maize comes up twice in 2017 (maize-soybean rotation and soybean-wheat/red clover-maize-soybean rotation) whereas soybean just comes up once in 2017 (Clover-maize-soybean).

Table 7 and Table 8 are the same!!! Decimals not needed-I apologize for that-my bad. I have moved Table 7 to the supplementary file and have labeled it Table S4. But I need to retain the decimals otherwise some of the costs below $0.50 would show up as zero and then the costs wouldn’t add up and reviewers or readers would think that I didn’t add the costs up correctly.

L415: Why presenting the rotation if rotation has no effect-good point-should have averaged across the rotations. Have amended.

L428: crops in 2016 are not organic!!!! It remains previous fertility-good point-have amended to read “organically-managed”.

L431-435: not clear-have deleted.

L440: “still do not use herbicides” => source??-have deleted.

L442-445: Why comparing high-conventional vs high-OA? What is the hypothesis beneath this? No hypothesis-these are complementary treatments-many mainstream farmers use high-input management (probably a majority disregard my recommendations and go with high input) and many organic famers use high seeding rates and apply lots of dairy manure that many of us think are too much (no laws on how much manure can be applied in my region).

L447: use of chicken manure is standard? No-not standard by most but it is used by organic growers who do not have access to solid dairy manure.

L485-487: “We also […] greatly.” => contradictory respect to the previous sentence. Have amended these two sentences to read as “. Consequently, we suggested that the best strategy for a transitioning grower to an organic cropping system in our environment may be to select soybean as the entry crop (instead of using a green manure crop) during the first year of transition followed by wheat/red clover because the returns would be the least negative [15]”.

 Table 10: standard deviations lacking-have added standard errors of the mean of the farming system x crop rotation x management input interaction.

L504-506: tests not presented in table 11 (only between rotations, not organic vs conventional)!! I have added the standard errors of the mean for the 3-way interaction so you can now get a feel for the variability. If you want, I can provide the ANOVA table that clearly shows that the p-value is significant for the 3-way interaction.

L504-506: tests not presented in table 11 (only between rotations, not organic vs conventional)!! I mentioned in the text that the HSD values were presented for the 2-way interactions-I am showing the farming system x input interaction when you compare the means going down the column of a specific rotation-the conventional high and the organic high is a valid comparison as is the conventional low and the organic low or the conventional low vs, the organic high and so forth.

L519-521: Results presented here are not significative-this discussion has been deleted.

L533: S-W/Cl-M-S has good partial returns only because 2015 and/or 2016 were favourable for the crops? Not really-more so because 2017 was so good for maize. Soybean did not yield well in 2015 and wheat did not yield well in 2016. 2017 maize yielded very well, much higher than 2015 and 2016 maize. 2018 soybean and wheat yielded much higher than 2015 soybean and much higher than 2016 wheat.

L542: + introduction of cover-crops ? No they had alfalfa in the rotation for 3 years that they harvested as hay-so they compared a 6-year organic rotation (M-S-W/A-A-A) with a 2 year conventional rotation (M-S).

L544-549: too much centred on an example. Have reduced the length of discussion on the study-but the authors in the cited study have a direct comparison with my study on a 2-year (maize-soybean) and 3-year (maize-soybean-wheat) rotation with the same crops so it is very relevant to my study.

Table 12: add standard deviations-have added standard errors.

L561: conventional neither responded to high input-good point-have added conventional.

L562-563: “Many organic growers […] weed control” => what is the data? Why not presented in context? Not sure what you are driving at here. I provided the maize reference that has all the weed density data in maize, my wheat reference for the weed density data in wheat, and the previous economics paper that had soybean weed density for the first 2 years of the study. I didn’t present it because I didn’t think that it fit for this manuscript.

L566: “first 4 years” => it is in fact transition 2 years conversion + 2 years OA-good point-I have amended.

L578-580: Not enough supported by the results presented. How to manage the introduction of soybean in first year of conversion to OA at the level of an entire farm? Not talking about the entire farm-I assume that growers will transition on a field by field basis and I specifically mentioned “we recommend that growers time their transition to a field where soybean is the intended crop”.

L582-586: it is very surprising, that crops or rotations can be suggested for an entire region. I doubt that the techniques presented here can be transposed identically on an entire region. See literature on rule-based cropping systems.-Good point but my job as an extension agronomist with 50% research responsibilities for 40 years was to do just that-make recommendations for a region. And like yourself, my recommendations were doubted by many-part of the reason why the seeding and N rates by many growers in my region are far above my recommendations for maize and wheat.  The 2019 Beltsville MD study recommended a 6-year organic rotation for the Mid-Atlantic USA region-and she didn’t even have an extension appointment but was rather a Federal researcher-an economist at that making recommendations for 4-5 states in the USA.

Reviewer 2 Report

Dear Authors,

The manuscript is well done. There are only minor remarks. In line 35 the correct latin name of red clover should be Trifolium pratense. In line 135 the S-W/Cl-M-S is met for the first time in chapter Materials and Methods. In line 229 the specification of W/Cl is given as 'frost-seeding'. As a reader unfamiliar with the practice 'frost-seeding', I could have liked to find this explanation in connection to line 35. W/Cl might be interpreted as alternatively, which is not much meaningful. For which reason is Table 8 (line 406-409) included? By searching the text for 'Table 8' there was only one hit and no reference to Table 8 in the text. As far as I can see the table caption as well as the figures of the that table are identical to those of Table 7.

Author Response

Response to Reviewer #2

Thank-you for your kind review.

In line 35 the correct latin name of red clover should be Trifolium pratense.-sorry about that-major mistake-have amended.

 In line 135 the S-W/Cl-M-S is met for the first time in chapter Materials and Methods. In line 229 the specification of W/Cl is given as 'frost-seeding'. As a reader unfamiliar with the practice 'frost-seeding', I could have liked to find this explanation in connection to line 35. W/Cl might be interpreted as alternatively, which is not much meaningful. Good point-I have amended to read “The clover in the S-W/Cl-M-S rotation was frost-seeded into a standing wheat crop at the end of winter when the soil was still frozen”.

For which reason is Table 8 (line 406-409) included? By searching the text for 'Table 8' there was only one hit and no reference to Table 8 in the text. As far as I can see the table caption as well as the figures of the that table are identical to those of Table 7.

My bad-I have moved the tables on costs to the supplementary fieles and have only provided one table on soybean costs because the costs were essentially the same between the 2017 and 2018 growing season (a few dollars different because of slightly higher seed and other input costs) but differences between soybean treatments were the same. I have added a sentence explain this in the revised submission.

Reviewer 3 Report

I think that the topic of the article fits the scope of the journal, and the readers of the journal will most likely find the study interesting.

The good in thing in the study is the experimental design. The 4 year split-split-split-plot experiment where 3 crop rotations and 2 cropping systems where studied is interesting and it has produced valuable data. Although, it is left unclear whether the plots where randomised to different cropping systems and crop rotations. This should be clearly written in the text.

However, there is a lot of room for improvement in the paper. First, statistical and economic analysis is not explained in sufficient clarity. In fact, those are not explained at all. Also, the statistical and economic models are not shown. Therefore, one cannot replicate the results. I would suggest the authors to explain in detail the statistical analysis and, more importantly in this study, the economic calculations. It would make the paper much easier to read if there would be some theoretical background provided in the materials and methods section. Now, I don't know what has been done in the paper.

The agoronmic aspects of the study are explained in a too detailed fashion, whereas the economic details are not explained at all. Thus, I suggest to focus more on the economics. Of course, this is just my suggestion. But I guess that if this paper is aimed to be understandable for readers who have background in agricultural economic, rather than agronomy, it should focus more on the economic content.

One thing that confused me almost throughout the paper, was the term "partial return".  How is it calculated? It is not explained in the text. In fact, any calculations are not explained in the text. The partial returns can be understood by looking the table 10 and doing the calculations by self, but it is frustrating that those are not explained at any point. If the authors were just used the term profit, or net return, I would have understood immediately what is compared in the study.  
I would suggest to use the term net returns (or profits) instead of partial returns. And clearly show how the economic calculations have been done.
It is a standard in economic analysis to compare profits (net returns) of the projects over the whole planning horizon. Thus, it would make sense to compare the profits of the different rotations and cropping systems. The results could be shown as figures, where annual profits where on y-axis and years on x-axis. Authors could also compare the cases where organic premium is added to the selling price, and where it is not added, to emphasise the meaning of the organic premium. Also, the net present values of the different rotations could be calculated, in order to compare clearly the management strategies. Alternatively, authors could compare the average annual net returns of the management strategies.
Also, annual yield development over the planning horizon (2015-2018) could be shown for different rotations and cropping systems.
I suggest that the authors focus on the whole planning horizon (2015-2018) in this study, because the authors have already studied the first 2 years in the previous study.
Language check is also recommended. There are sentences that make no sense. It gives an impressions that the study is written in a hurry. There are way too much numbers and higher/lower comparisons in the text. It is not written in a reader-friendly fashion, although it could be, because the content is not too difficult in a theoretical sense; the content of the paper is not so difficult, but the message gets lost behind all the details. I suggest that the text is rewritten so that it is easer to read. Also, the structure of the paper is messy (particularly in the results section).

I would also suggest to add some general economis crop rotation studies in the literature review section, to put the study in a wider context. I added some suggestions in the comment. Authores could look at those studies, or some other similar studies, and formulate their model in a similar fashion.

In conclusion, the main suggestions are that the authors explain in detail the statistical and economic analysis in method section. In results the authors could focus on showing the development of the annual net returns of different crop rotations and cropping systems, and use figures instead of tables to illustrate the results. Also the net present values over the whole 4 year planning horizon could be shown, because those are the best way to compare different management strategies. I understand that the economic comparison of the organic and conventional cropping systems, and different rotations, is, after all, the aim of the paper.

I have some more detailed comments in the attached file.

Author Response

Response to Reviewer #3

Thank-you for your thoughtful and thorough review-below are my responses to your comments and suggestions.

The good in thing in the study is the experimental design. The 4 year split-split-split-plot experiment where 3 crop rotations and 2 cropping systems where studied is interesting and it has produced valuable data. Although, it is left unclear whether the plots where randomised to different cropping systems and crop rotations. This should be clearly written in the text. Good point-I have amended to read:  “The experimental design was a randomized block in a split-split-split plot arrangement with four replications.”

First, statistical and economic analysis is not explained in sufficient clarity. In fact, those are not explained at all. Also, the statistical and economic models are not shown. Therefore, one cannot replicate the results. I would suggest the authors to explain in detail the statistical analysis and, more importantly in this study, the economic calculations. It would make the paper much easier to read if there would be some theoretical background provided in the materials and methods section. Now, I don't know what has been done in the paper. I don’t understand how you can say this-the text reads: “Previous crop (or fields), farming systems (conventional and organic), crop rotations ((Cl-M-S-W; M-S-M-S; and S-W/Cl-M-S), and management inputs (recommended and high) were considered fixed and replications random for statistical analyses of yield, revenue, and returns for each crop within individual years, using the REML function in the MIXED procedure of SAS (version 9.4; SAS Institute Inc., Cary, NC, USA). We also used the same SAS procedures to statistically analyze the revenue and partial returns of the three rotations for the 2-year transition period, the first 2-years after transition when organic crops are eligible for the organic premium, and the entire 4-year study period. We could not statistically analyze costs because the same inputs were applied to the four replications so there was no random variability and thus no statistical analysis. Fields with different previous crops (2014 crops) had yield differences for maize and soybean in 2015 but did not have any main effects or interactions with farming systems, rotations, or management inputs in subsequent years or in the 2-year or 4-year economic analyses. Consequently, the data will be pooled across previous crops (the three contiguous fields) for the 2017 and 2018 years, and across the 2-year and 4-year data sets. Least square means of the main effects (cropping system, crop rotations, and management inputs) were computed and means separations were performed on significant effects using Tukey’s HSD (Studentized Range) test, with statistical significance set at p < 0.05. Differences among least square means for farming system interactions were calculated also using Tukey’s HSD test. Two-way and three-way interactions were detected for most variables so the interaction”.

I have devoted an entire paragraph to our statistical procedures-this is an empirical agronomic data and this is how we analyze our experiments. There is also a paragraph devoted to how we did the economics-

Economic analyses focused on enterprise budget items that differed among the treatments, namely the value of production associated with yield differences as well as cost differences for inputs for maize, soybean and wheat. Returns above variable and fixed input costs that do not differ between conventional and organic soybean production under recommended and high input management were calculated for the three crops. Our selected variable inputs include: seed, fertilizer, and sprays/other inputs (herbicide, fungicide, inoculant, and organic seed treatment); labor and machinery operating inputs (repairs and maintenance, fuels and lubricants), excluding tillage, planting and harvesting tasks, except for hauling, where hauling cost is a function of yield [18]. Cost of production values reported for fixed inputs exclude farm machinery ownership costs for tillage, planting and harvest, land charges, and values of management inputs. Grain drying is not required for soybean and wheat. Grain moistures did not differ between organic and conventional maize, so we did not include those production costs in maize”.

I think that part of the problem was the term “partial budget analyses”-this is a common analyses done for farm business decisions in the USA-I have pasted how the Department of Agricultural Economics at Iowa State describes partial budgeting.

Partial Budgeting: A Tool to Analyze Farm Business Changes

Business owners must often make decisions about changes they are either contemplating making or that have to be made. Many of the decisions are incremental, such as adding land, expanding or reducing an enterprise or changing how an enterprise is managed. The partial budget is a useful tool for farm managers when these situations arise. A partial budget helps farm owners/managers evaluate the financial effect of incremental changes.

A partial budget only includes resources that will be changed. It does not consider the resources in the business that are left unchanged. Only the change under consideration is evaluated for its ability to increase or decrease income in the farm business.

The agoronmic aspects of the study are explained in a too detailed fashion, whereas the economic details are not explained at all. Thus, I suggest to focus more on the economics. Of course, this is just my suggestion. But I guess that if this paper is aimed to be understandable for readers who have background in agricultural economic, rather than agronomy, it should focus more on the economic content-good suggestion and have greatly reduced the agronomic discussion in the revised manuscript. In fact, I have greatly reduced the entire manuscript based on many of your presentations. I am afraid that I am too data-driven and it shows up in my writing and speaking for that matter. I have shortened the manuscript significantly by reducing the Introduction by 27% (from 1277 down to 887 words-there I go again providing data!), M&M section by 43% (2206 words down to 1257 words), the discussion on the individual crops by 62% (1863 to 710 words), and the discussion on the crop rotations by 18% (1087 to 877 words). 

One thing that confused me almost throughout the paper, was the term "partial return".  How is it calculated? It is not explained in the text. In fact, any calculations are not explained in the text. The partial returns can be understood by looking the table 10 and doing the calculations by self, but it is frustrating that those are not explained at any point. If the authors were just used the term profit, or net return, I would have understood immediately what is compared in the study. I would suggest to use the term net returns (or profits) instead of partial returns. And clearly show how the economic calculations have been done.  Sorry about that-I assumed that this was common knowledge for all. I have amended partial returns to read “returns above selected production costs” or just returns to allay your confusion and possibly that of other readers. Again, I thought that I spelled out pretty clearly how it was calculated-the amended version now reads: “Returns above variable and fixed input costs that do not differ between conventional and organic soybean production under recommended and high input management were calculated for the three crops. Our selected variable inputs include: seed, fertilizer, and sprays/other inputs (herbicide, fungicide, inoculant, and organic seed treatment); labor and machinery operating inputs (repairs and maintenance, fuels and lubricants), excluding tillage, planting and harvesting tasks, except for hauling, where hauling cost is a function of yield [18]. Cost of production values reported for fixed inputs exclude farm machinery ownership costs for tillage, planting and harvest, land charges, and values of management inputs”.

It is a standard in economic analysis to compare profits (net returns) of the projects over the whole planning horizon. Thus, it would make sense to compare the profits of the different rotations and cropping systems. The results could be shown as figures, where annual profits where on y-axis and years on x-axis. Authors could also compare the cases where organic premium is added to the selling price, and where it is not added, to emphasise the meaning of the organic premium. Also, the net present values of the different rotations could be calculated, in order to compare clearly the management strategies. Alternatively, authors could compare the average annual net returns of the management strategies.  The returns above selected production costs or “net returns” have been clearly shown in Tables 6-8 (in the revision-all the tables with costs have been moved to the supplementary files). I do not like figures for manuscripts because I never know the exact data-I can’t tell what the differences are in figures, whereas the exact values are in the Table. Of course, if I was presenting this topic at a meeting I would use figures but for manuscripts I prefer the preciseness of the tables that clearly shows the data.

Also, annual yield development over the planning horizon (2015-2018) could be shown for different rotations and cropping systems. I am a production agronomist who can do simple production economics (i.e partial budgeting and enterprise budgets). I just conducted an empirical field experiment that includes economics along with the yield data.  I have written this manuscript for like-minded people whom I presume are agronomists reading the journal, Agronomy-I did not submit this to an Agricultural Economics journal (although the first 2 years of the data was selected as an invited presentation for the International Agricultural Economics triennial meetings in Vancouver last year-and yes, I used figures).

I suggest that the authors focus on the whole planning horizon (2015-2018) in this study, because the authors have already studied the first 2 years in the previous study.-Good point-but I have to mention the first 2 years in order to discuss the 4 years.

Language check is also recommended. There are sentences that make no sense. It gives an impressions that the study is written in a hurry. There are way too much numbers and higher/lower comparisons in the text. It is not written in a reader-friendly fashion, although it could be, because the content is not too difficult in a theoretical sense; the content of the paper is not so difficult, but the message gets lost behind all the details. I suggest that the text is rewritten so that it is easer to read. Also, the structure of the paper is messy (particularly in the results section). Sorry about that-I can assure you that the manuscript was not written in a hurry-it took me 6 full weeks to write this. I am retired now so I devoted most of the day to it so it took a lot of effort and time to put this together. But I have been retired 2.5 years now and 71 years old so perhaps I am slipping a bit-I hope that you find the revision much easier to read. I am data-driven and have trouble writing or speaking about anything without data. I think that it is important for future readers who will spend more time with it, rather than just reviewing it, to have all the data at their disposal.

I would also suggest to add some general economis crop rotation studies in the literature review section, to put the study in a wider context. I added some suggestions in the comment. Authores could look at those studies, or some other similar studies, and formulate their model in a similar fashion. I guess that I disagree. I think that I should focus on papers that have similar crops and crop rotations-in my case the maize-soybean-maize-soybean or the soybean-wheat/red clover-maize-soybean rotation. I don’t think that it makes any sense to discuss crop rotations with different crops, such as rice, cotton, potatoes, or vegetable crops, just because it is a crop rotation study. When I published seeding rate studies on maize, I did not discuss the literature on wheat seeding rates. Again, this was an empirical agronomic study by a production agronomist that utilized economics to explain the empirical results of a 4-year study comparing maize, soybean, and wheat in 3 different crop rotations under conventional and organic farming systems.

In conclusion, the main suggestions are that the authors explain in detail the statistical and economic analysis in method section. In results the authors could focus on showing the development of the annual net returns of different crop rotations and cropping systems, and use figures instead of tables to illustrate the results. Also the net present values over the whole 4 year planning horizon could be shown, because those are the best way to compare different management strategies. I understand that the economic comparison of the organic and conventional cropping systems, and different rotations, is, after all, the aim of the paper. I don’t think that I can improve upon the discussion of the statistical analyses of the study-this is routine Mixed Model analyses that agronomists use and this is how we describe what we do. The economics is simple addition and subtraction (all I am capable of doing!)-we just added up the production costs for each farming system that would change if transitioning from conventional to organic, multiply the yields by the prices for the commodity (with the organic premium for the organic crops in the last 2 years of the study), and subtract the selected production costs from the revenue to come up with the returns and then run the returns through our statistical analyses to see which farming system was most economical. I show the development and the annual returns above selected production costs for the transition years, the first 2 years after certification, and for the full 4-years-I prefer table presentation instead of figure presentation-I am always disappointed when there are figures instead of tables in the literature and I can’t get the exact difference between the comparisons.

Response to comments on the manuscript: (some of your highlights did not have comments)

Abstract: Unfortunately you didn’t understand what “partial returns” meant-again it is a typical term used by production agronomists and agricultural economists. I had a graduate student TA the class in 2012 and he told me that it still a staple in the Farm Business Management class that I took in the early 1970s.  I must have published at least 5 but maybe more papers using this terminology. But I have changed the term to “returns above selected costs” to insure that other readers do not miss this point. But the abstract has a 200 word guideline so I have used “returns above selected costs” when if first mention returns in the abstract but leave is as “returns” at all other times.

Also, I have reworded the sentences saying higher returns despite lower yields and higher productions costs to include “because of the organic price premium”.

Introduction-“literature is too detailed-we do not need all these numbers”. I have shortened the Introduction by about 27% and have rephrased to emphasize the crop rotations. I agree that there was too much data in the Introduction but I believe in providing data in the Introduction (ironically because I was criticized by reviewers early in my career for not providing data in the Introduction and just saying “higher” or “lower”). So I have provided data in the comparisons that I have retained in the manuscript.

Provide more information focusing on the economics of crop rotation ? I have emphasized the crop rotation and not the individual crops in the revised manuscript (except for the USDA survey on specific crops because they did not provide rotation data for maize, soybean, and wheat. But I believe that the rotations that I discuss need to be specific to the study-not just rotations in general. So I cited all studies that I could find that compared conventional and organic maize-soybean and maize-soybean-wheat/red clover rotations. I don’t think that I should discuss rotations of different crops even if the study compared conventional and organic rotations with different crops. Likewise, I don’t think that I should compare rotations that are not specifically comparing conventional and organic rotation. I didn’t even include my own crop rotation studies that compared maize-soybean and maize-soybean-wheat/red clover rotations from the 1990s because they did not compare these rotations under conventional and organic farming systems. (Katsvairo, T. W., Cox, W. J. (2000). Economics of cropping systems featuring different tillage, rotation, and management systems. Agronomy Journal, 92, 485-493; Singer, J. W., Cox, W. J. (1998). Economics of different crop rotations in New York. Journal of Production Agriculture, 11, 447-451).

L. 54, 67, 71, etc. I used the exact economic terms that the authors used in my literature review so sometimes it is net present value, sometimes it is it net revenue, sometimes it is economic mean return. There seems to be lots of terms that can be used so I am disappointed that my economic term was so misunderstood.

L.86 “Yielded 69% of”-again that this is the exact term used by USDA economists-the authors were not agronomists but agricultural economists who used this wording that confused you.

L. 100-108. As suggested by Reviewer #1, I have eliminated all the data in this section because he or she said it was outdated-I would have preferred to have kept it in but decided to eliminate.

L.119-focus on the 4-year study. I have 1 table and one paragraph that addresses material from the previous manuscript. I don’t see how I can discuss the 4-years of data and ignore the first 2 years. I have opted to greatly reduce the discussion on the previous data but feel that is integral to the whole story so have provided some discussion on it.

L. 142 Yes-the experiment was randomized and have rewritten the sentence to implicitly state it.

L. 159 Yes-it was no-till as clearly defined on line 69 of the original manuscript.

Line 168-185. Have rewritten the entire section and have shortened a great deal and have referred to the previous study for more detail and the table.

Table 1. I have eliminated the soil texture data from the old Table 1 (now Table 2). It is the exact same soil for both systems because they are randomized in four replications across the three fields.

L. 167-186. Have rewritten.

L.274-more description of the statistics. You are obviously not familiar with the way that agronomists analyze their studies. There is sufficient information for anyone who runs agronomic studies to know what we did. If you peruse through any articles in this journal or other Agronomy Journals you will see that I have provided more than the normal amount of information. You told me that I provided too much information on agronomic methods but now you are asking for more information on the statistics. We obviously have different priorities on what it important. I am an agronomist writing an article for the journal, Agronomy. I think that most agronomists will totally understand how we analyzed the data statistically.

L.275-276. I am not sure what you are looking for here-there is no statistical or economic model. This is strictly a routine empirical agronomic study in a randomized complete block design analyzed statistically as we do with yield. Likewise, the revenue and returns above selected costs were just plugged into the statistical analysis just like the yield data was plugged in. It is an empirical study analyzed statistically with no economic model. Just the returns above selected production costs analyzed as we analyze yield data.

L. 288 Results and Discussion-I have rewritten this section and shortened it by ~38%, including 62% on individual crops. But I believe that you have to first discuss the crops before you discuss the crop rotations-you have to know what went into the economics of the crop rotation before you can discuss it. I have followed the example of other manuscripts including the recent one published in Agronomy Journal this year, written by an agricultural economist. She discussed costs of each crop first, yield second, and net return to the cropping system last. (White, K.E., M.A. Cavigelli, Conklin, A.E., Rasman, C. Economic Performance of Long-term Organic and Conventional Crop Rotations in the Mid-Atlantic. Agron. J. 2019, 111, 1-3. Doi: 10.2134/agronj2018.09.0604). You can see how she analyzed the economic data just like she analyzed yield-no economic model.

I don’t consider the yield, revenue, costs and returns above selected costs minor details. I believe that you need that data to understand the farming system data. I prefer to use tables with actual data that the reader can see rather than figures that the reader must guess at what the actual values are. If I were giving a presentation, I would obviously use figures. But in a manuscript, I assume that the readers who are most interested in the topic, not the casual reader but those who are conducting similar work, will take the time to study the data to see how it relates to their study.

The Results and Discussion section has been reduced by 38% in length so some of the data that you find unimportant are no longer in the text (but remain in the Tables).

Table 3. I don’t agree that the yield data is unimportant and only the returns above selected costs are important. Again, I am an agronomist and believe yield, costs, revenue, and returns are all important. You are an economist who doesn’t have an interest in the yield data. I believe that most the readers of this journal are agronomists so I think that we need to leave the yield and revenue data in the table.

Table 4-I have moved to the supplementary file.

Yes-yield penalty means lower yields-again a term used by other agronomists:

Table 5-you want to see the net revenues from 2015 and 2016? The entire manuscript you have been asking me to not show the data from the transition years (2015 and 2016) but rather to concentrate on the 2 years after the transition and the 4 years of the study? It is in the previous manuscript for the individual crops but is in Table 6 in the revision for the 3 crop rotations.

Table 6-you want to see the net revenues from the other years also? Again the net revenues for maize from the other years is in the previous manuscript. I am confused now on just want you want-this is getting very discouraging.

Tables 7 and 8 are identical-you are right. I must be losing it or slipping pretty badly to have done that. Sorry about that.

Table 10-Explain what it meant by partial returns in the M&M section. I have writtenReturns to variable and fixed inputs that do not differ between conventional and organic production under recommended and high input management were calculated for the three crops. Our selected variable inputs include: seed, fertilizer, and sprays/other inputs (herbicide, fungicide, inoculant, and organic seed treatment); labor and machinery operating inputs (repairs and maintenance, fuels and lubricants), excluding tillage, planting and harvesting tasks, except for hauling, where hauling cost is a function of yield [18]. Cost of production values reported for fixed inputs exclude farm machinery ownership costs for tillage, planting and harvest, land charges, and values of management inputs.

Table 11-I think that it is important to show how returns above selected costs were derived-from the costs and revenues.

Table 12-why would a figure make it easier to see and compare. How could it be easier to see and compare than this:

CONVENTIONAL

Recommended

1992 b

High Input

1605 c

ORGANIC

Recommended

3012 a

High Input

1993 b

The actual data is right there with lettering indicating what treatments are different. How could a figure possibly make it any clearer?  We must see things differently. Or perhaps you are younger and prefer to look at data differently-the main thing that I am trying to show is the comparison between organic and conventional systems with high and low inputs and to me it is far clearer and far more exact in table format than any picture could give.

Conclusions: Good suggestions-have incorporated “Results indicate” and have added a sentence on the limitations of the study.

I am sorry that your review of the original submission of the manuscript was such a frustrating experience for you. The manuscript has been totally revised-hopefully you find the revision greatly improved.

Round 2

Reviewer 1 Report

The manuscript was largely rewritten and I think the global message of the paper is much clearer now.

All the remarks made were taken into account and, when the authors disagree, justifications are made. 

Author Response

Thank-you.

Reviewer 3 Report

I think that the manuscript has been significantly improved and now need minor improvements or as it is.

Author has not followed the suggestion that the abstract should be rewritten so that the main message would be clear, and removing all the unnecessary numeric details. Perhaps, however, it is suitable in this context because the paper does not try to produce any general results.
Author has also ignored the suggestion that the statistical methods should be clearly described. The author refers to the statistical commands in a particular software. The reader should be able to replicate the results with any software. I think it should be explained what exactly is the REML function and MIXED procedure. I should be able to obtain same results by using R, Stata or Matlab. But again, perhaps this is sufficient reporting in this context, because the statistical analysis is not the main contribution of the paper, although I am aware that in this journal statical methods are sometimes much more deeply described. But of course it depends on the general style of the journal what is the needed level of detail in method section. Perhaps agronomist using SAS understand immediately what has been done here. Perhaps you could give more detailed description in the supplementary material? But it may also be unnecessar.

The results section is now much clearer and easier to follow, and also the conclusions. The text is not written in a way that would be easy to follow for a person not coming from an agronomic background, but as the author points out, this is agronomy journal and as such targeted to specialised readers.

Author Response

Author has not followed the suggestion that the abstract should be rewritten so that the main message would be clear, and removing all the unnecessary numeric details. Perhaps, however, it is suitable in this context because the paper does not try to produce any general results.

I have followed the guidelines of Agronomy Journal on how to write an abstract (see their example in their Style Manual below). They clearly want data in the Abstract:

Dryland Grain Sorghum Water Use, Light Interception, and Growth Responses to Planting Geometry J. L. Steiner* 

ABSTRACT 

Rationale Crop yields are primarily water-limited under dryland production systems in semiarid regions. Objectives or This study was conducted to determine whether the growing-season water hypothesis balance could be manipulated through planting geometry. 

Methods The effects of row spacing, row direction, and plant population on the water use, light interception, and growth of grain sorghum [Sorghum bicolor (L.) Moench] were investigated at Bushland, TX, on a Pullman clay loam (fine, mixed, superactive thermic Torrertic Paleustoll).Results In 1983, which had a dry growing season, narrow-row spacing and higher population increased seasonal evapotranspiration (ET) by 7 and 9%, respectively, and shifted the partitioning of ET to the vegetative period. Medium population crops yielded 6.2 and 2.3 Mg ha–1 of dry matter and grain, respectively. High population resulted in high dry matter (6.1 Mg ha–1) and low grain yield (1.6 Mg ha–1), whereas low population resulted in low dry matter (5.4 Mg ha–1) and high grain yield (2.3 Mg ha–1). Row direction did not affect water use or yield. In 1984, dry matter production for a given amount of ET and light interception was higher in the narrow-row crops. Evapotranspiration was less for a given amount of light interception in the narrow-row crops and in the north–south row crops. 

Conclusions Narrow-row planting geometry appears to increase the partitioning of ET to the transpiration component and may improve the efficiency of dryland cropping systems.

Author has also ignored the suggestion that the statistical methods should be clearly described. The author refers to the statistical commands in a particular software. The reader should be able to replicate the results with any software. I think it should be explained what exactly is the REML function and MIXED procedure. I should be able to obtain same results by using R, Stata or Matlab. But again, perhaps this is sufficient reporting in this context, because the statistical analysis is not the main contribution of the paper, although I am aware that in this journal statical methods are sometimes much more deeply described. But of course it depends on the general style of the journal what is the needed level of detail in method section. Perhaps agronomist using SAS understand immediately what has been done here. Perhaps you could give more detailed description in the supplementary material? But it may also be unnecessary.

I have clearly stated that previous crops, farming systems, crop rotations, and management inputs were fixed variables and replications were random variables in using the REML Function of the MIXED Model. I have never seen anyone write their statement options when using SAS.

"

MODEL Statement

MODEL dependent = </ options> ;

The MODEL statement names a single dependent variable and the fixed effects, which determine the  matrix of the mixed model (see the section Parameterization of Mixed Models for details). The specification of effects is the same as in the GLM procedure; however, unlike PROC GLM, you do not specify random effects in the MODEL statement. The MODEL statement is required.

And I suppose that I could add the basic features of the MIXED model and REML functions (below) but I really don't think that it will add much to most readers.

PROC MIXED fits the specified mixed linear model and produces appropriate statistics. Here are some basic features of PROC MIXED:  covariance structures, including variance components, compound symmetry, unstructured, AR(1), Toeplitz, spatial, general linear, and factor analytic  GLM-type grammar, by using MODEL, RANDOM, and REPEATED statements for model specification and CONTRAST, ESTIMATE, and LSMEANS statements for inferences  appropriate standard errors for all specified estimable linear combinations of fixed and random effects, and corresponding t and F tests  subject and group effects that enable blocking and heterogeneity, respectively  REML and ML estimation methods implemented with a Newton-Raphson algorithm  

The results section is now much clearer and easier to follow, and also the conclusions. The text is not written in a way that would be easy to follow for a person not coming from an agronomic background, but as the author points out, this is agronomy journal and as such targeted to specialised readers. 

I agree.